# Development of a polygenic risk score to improve screening for fracture risk: A genetic risk prediction study

Vincenzo Forgetta[1◉], Julyan Keller-Baruch[2◉], Marie Forest[1], Audrey Durand[3], Sahir Bhatnagar[1], John P. Kemp[4,5], Maria Nethander[6,7], Daniel Evans[8], John A. Morris[1], Douglas P. Kiel[9], Fernando Rivadeneira[10], Helena Johansson[11,12], Nicholas C. Harvey[13,14], Dan Mellström[7], Magnus Karlsson[15], Cyrus Cooper[13,14,16], David M. Evans[4,5], Robert Clarke[17], John A. Kanis[11,12], Eric Orwoll[18,19], Eugene V. McCloskey[20], Claes Ohlsson[7], Joelle Pineau[3], William D. Leslie[21], Celia M. T. Greenwood[1,2,22,23], J. Brent Richards[1,2,24] *

1 Centre for Clinical Epidemiology, Department of Medicine, Lady Davis Institute, Jewish General Hospital, McGill University, Montréal, Québec, Canada, 2 Department of Human Genetics, McGill University, Montréal, Québec, Canada, 3 School of Computer Science, McGill University, Montréal, Québec, Canada, 4 University of Queensland Diamantina Institute, University of Queensland, Woolloongabba, Queensland, Australia, 5 Medical Research Council Integrative Epidemiology Unit, Population Health Sciences, Bristol Medical School, University of Bristol, Bristol, United Kingdom, 6 Bioinformatics Core Facility, Sahlgrenska Academy, University of Gothenburg, Gothenburg, Sweden, 7 Centre for Bone and Arthritis Research, Department of Internal Medicine and Clinical Nutrition, Institute for Medicine, Sahlgrenska Academy, University of Gothenburg, Gothenburg, Sweden, 8 California Pacific Medical Center Research Institute, San Francisco, California, United States of America, 9 Institute for Aging Research, Hebrew SeniorLife, Department of Medicine, Beth Israel Deaconess Medical Center and Harvard Medical School, Broad Institute of MIT & Harvard University, Boston, Massachusetts, United States of America, 10 Department of Internal Medicine, Erasmus Medical Center, Rotterdam, Netherlands, 11 Centre for Metabolic Bone Diseases, University of Sheffield, Sheffield, United Kingdom, 12 Australian Catholic University, Melbourne, Victoria, Australia, 13 Medical Research Council Lifecourse Epidemiology Unit, University of Southampton, Southampton, United Kingdom, 14 National Institute for Health Research Southampton Biomedical Research Centre, University of Southampton and University Hospital Southampton NHS Foundation Trust, Southampton, United Kingdom, 15 Department of Orthopaedics and Clinical Sciences, Lund University, Skane University Hospital, Malmö, Sweden, 16 National Institute for Health Research Oxford Biomedical Research Centre, University of Oxford, Oxford, United Kingdom, 17 Clinical Trial Service Unit and Epidemiological Studies Unit, University of Oxford, Oxford, United Kingdom, 18 Bone and Mineral Unit, Oregon Health & Science University, Portland, Oregon, United States of America, 19 Department of Medicine, Oregon Health & Science University, Portland, Oregon, United States of America, 20 Mellanby Centre for Bone Research, Centre for Integrated Research in Musculoskeletal Ageing, University of Sheffield and Sheffield Teaching Hospitals Foundation Trust, Sheffield, United Kingdom, 21 Department of Medicine, University of Manitoba, Winnipeg, Manitoba, Canada, 22 Gerald Bronfman Department of Oncology, McGill University, Montréal, Québec, Canada, 23 Department of Epidemiology, Biostatistics & Occupational Health, McGill University, Montréal, Québec, Canada, 24 Department of Twin Research and Genetic Epidemiology, King's College London, London, United Kingdom

◉ These authors contributed equally to this work.

* brent.richards@mcgill.ca



**Data Availability Statement:** All relevant summary-level data are within the manuscript and

## Abstract

### Background

Since screening programs identify only a small proportion of the population as eligible for an intervention, genomic prediction of heritable risk factors could decrease the number needing to be screened by removing individuals at low genetic risk. We therefore tested whether a

its Supporting Information files. All other relevant underlying individual-level data will be returned to UK Biobank in accordance with the signed Material Transfer Agreement. UK Biobank will then make this individual-level data available researchers in accordance with their data access policies. UK Biobank can be contacted by email at access@ukbiobank.ac.uk.

**Funding:** This program was funded by the Canadian Institutes of Health Research. UK Biobank is funded by the Wellcome Trust, UK Medical Research Council, Department of Health, Scottish Government and the Northwest Regional Development Agency. It has also had funding from the Welsh Assembly Government and the British Heart Foundation. None of these funders had a role in the design, implementation or interpretation of this study. The Richards lab is supported by the Canadian Institutes of Health Research, the Canadian Foundation for Innovation, the Lady Davis Institute and the Fonds de Recherche Santé Québec (FRSQ). Dr. Richards is supported by a FRQS Clinical Research Scholarship. TwinsUK is funded by the Wellcome Trust, Medical Research Council, European Union, the National Institute for Health Research (NIHR)-funded BioResource, Clinical Research Facility and Biomedical Research Centre based at Guy's and St Thomas' NHS Foundation Trust in partnership with King's College London. J.P.K is funded by a University of Queensland Development Fellowship (UQFEL1718945), and a National Health and Medical Research Council (Australia) Investigator grant (GNT1177938). CLSA is funded by the Canadian Institutes of Health Research and the Canadian Foundation for Innovation. MrOS: The Osteoporotic Fractures in Men (MrOS) Study is supported by National Institutes of Health funding. The following institutes provide support: The National Institute on Aging (NIA), the National Institute of Arthritis and Musculoskeletal and Skin Diseases (NIAMS), the National Center for Advancing Translational Sciences (NCATS), and NIH Roadmap for Medical Research under the following grant numbers: U01 AG027810, U01 AG042124, U01 AG042139, U01 AG042140, U01 AG042143, U01 AG042145, U01 AG042168, U01 AR066160, and UL1 TR000128. NIAMS provided funding for the MrOS ancillary study 'Replication of candidate gene associations and bone strength phenotype in MrOS' under the grant number R01 AR051124 and the MrOS ancillary study 'GWAS in MrOS and SOF' under the grant number RC2 AR058973. Dr. Nielson is supported by a K01 from NIAMS (K01AR062655). SOF: The Study of Osteoporotic Fractures (SOF) is supported by National Institutes of Health funding. The National

polygenic risk score for heel quantitative ultrasound speed of sound (SOS)—a heritable risk factor for osteoporotic fracture—can identify low-risk individuals who can safely be excluded from a fracture risk screening program.

## Methods and findings

A polygenic risk score for SOS was trained and selected in 2 separate subsets of UK Biobank (comprising 341,449 and 5,335 individuals). The top-performing prediction model was termed "gSOS", and its utility in fracture risk screening was tested in 5 validation cohorts using the National Osteoporosis Guideline Group clinical guidelines ($N$ = 10,522 eligible participants). All individuals were genome-wide genotyped and had measured fracture risk factors. Across the 5 cohorts, the average age ranged from 57 to 75 years, and 54% of studied individuals were women. The main outcomes were the sensitivity and specificity to correctly identify individuals requiring treatment with and without genetic prescreening. The reference standard was a bone mineral density (BMD)–based Fracture Risk Assessment Tool (FRAX) score. The secondary outcomes were the proportions of the screened population requiring clinical-risk-factor-based FRAX (CRF-FRAX) screening and BMD-based FRAX (BMD-FRAX) screening. gSOS was strongly correlated with measured SOS ($r^2$ = 23.2%, 95% CI 22.7% to 23.7%). Without genetic prescreening, guideline recommendations achieved a sensitivity and specificity for correct treatment assignment of 99.6% and 97.1%, respectively, in the validation cohorts. However, 81% of the population required CRF-FRAX tests, and 37% required BMD-FRAX tests to achieve this accuracy. Using gSOS in prescreening and limiting further assessment to those with a low gSOS resulted in small changes to the sensitivity and specificity (93.4% and 98.5%, respectively), but the proportions of individuals requiring CRF-FRAX tests and BMD-FRAX tests were reduced by 37% and 41%, respectively. Study limitations include a reliance on cohorts of predominantly European ethnicity and use of a proxy of fracture risk.

## Conclusions

Our results suggest that the use of a polygenic risk score in fracture risk screening could decrease the number of individuals requiring screening tests, including BMD measurement, while maintaining a high sensitivity and specificity to identify individuals who should be recommended an intervention.

## Author summary

### Why was this study done?

- Osteoporosis screening identifies only a small proportion of the screened population to be eligible for intervention.

- The prediction of heritable risk factors using polygenic risk scores could decrease the number of screened individuals by reassuring those with low genetic risk.

- We investigated whether the genetic prediction of heel quantitative ultrasound speed of sound (SOS)—a heritable risk factor for osteoporotic fracture—could be incorporated

Institute on Aging (NIA) provides support under the following grant numbers: R01 AG005407, R01 AR35582, R01 AR35583, R01 AR35584, R01 AG005394, R01 AG027574, and R01 AG027576. The National Institute of Arthritis and Musculoskeletal and Skin Diseases (NIAMS) provides funding for the SOF ancillary study 'GWAS in MrOS and SOF' under the grant number RC2AR058973. No funders had any influence over the study design, its implementation or interpretation.

**Competing interests:** I have read the journal's policy and the authors of this manuscript have the following competing interests: JAK reports grants from Amgen, Eli Lilly and Radius Health; consulting fees from Theramex. JAK is the architect of FRAX but has no financial interest. JBR reports investigator-initiated grants from Biogen, Eli Lilly and GlaxoSmithKline, for programs unrelated to the research presented here. JBR is an advisor to GlaxoSmithKline. DPK reports grants from Radius Health and the Dairy Council unrelated to the research presented here and consulting fees from Solarea Bio unrelated to the research presented here. CC reports personal fees (outside the submitted work) from Amgen, Danone, Eli Lilly, GSK, Kyowa Kirin, Medtronic, Merck, Nestle, Novartis, Pfizer, Roche, Servier, Shire, Takeda, UCB. NCH reports consultancy, lecture fees and honoraria (outside the submitted work) from Alliance for Better Bone Health, AMGEN, MSD, Eli Lilly, Servier, Shire, UCB, Kyowa Kirin, Consilient Healthcare, Radius Health and Internis Pharma.

**Abbreviations:** BMD, bone mineral density; BMD-FRAX, bone-mineral-density-based Fracture Risk Assessment Tool; CLSA, Canadian Longitudinal Study on Aging; CRF-FRAX, clinical-risk-factor-based Fracture Risk Assessment Tool; FRAX, Fracture Risk Assessment Tool; GWAS, genome-wide association study; LASSO, least absolute shrinkage and selection operator; NOGG, National Osteoporosis Guideline Group; SOF, Study of Osteoporotic Fractures; SOS, speed of sound.

into an established screening guideline to identify individuals at low risk for osteoporosis.

## What did the researchers do and find?

- Using UK Biobank, we developed a polygenic risk score (gSOS) consisting of 21,717 genetic variants that was strongly correlated with SOS ($r^2 = 23.2\%$).

- Using the National Osteoporosis Guideline Group clinical assessment guidelines in 5 validation cohorts, we estimate that reassuring individuals with a high gSOS, rather than doing further assessments, could reduce the number of clinical-risk-factor-based Fracture Risk Assessment Tool (FRAX) tests and bone-density-measurement-based FRAX tests by 37% and 41%, respectively, while maintaining a high sensitivity and specificity to identify individuals who should be recommended an intervention.

## What do these findings mean?

- We show that genetic pre-screening could reduce the number of screening tests needed to identify individuals at risk of osteoporotic fractures.

- Therefore, the potential exists to improve the efficiency of osteoporosis screening programs without large losses in sensitivity or specificity to identify individuals who should receive an intervention.

- Further translational studies are needed to test the clinical applications of this polygenic risk score; however, our work shows how such scores could be tested in the clinic.

## Introduction

Screening programs are generally designed to identify a proportion of the screened population whose risk of a clinically relevant outcome is high enough to merit an intervention. However, usually only a small proportion of individuals who undergo screening is found to be at high risk, indicating that much of the screening expenditure is spent on individuals who will not qualify for intervention.

Osteoporosis is a common and costly disease that results in an increased predisposition to fractures [1]. Many guidelines [2–6] aimed at the prevention of osteoporosis-related fractures incorporate the Fracture Risk Assessment Tool (FRAX) [7,8], a validated method to risk stratify individuals for treatment by assessing their 10-year probability of major osteoporotic fracture. Guidelines vary widely, but often recommend a staged process where individuals are first assessed with a clinical-risk-factor-based FRAX (CRF-FRAX), and those at increased risk of fracture are then additionally characterized with a more expensive bone mineral density (BMD)–based FRAX (BMD-FRAX) score. Such approaches are usually recommended in the setting of enhanced case-finding strategies, but recently, a large randomized controlled trial (SCOOP) demonstrated the potential benefit of community-based fracture risk assessment in reducing rates of hip fractures in elderly women [9]. This trial used a strategy based on the

National Osteoporosis Guideline Group (NOGG) screening strategy [3], which implements fracture risk stratification through the use of FRAX scores. In this trial, the entire screened population underwent FRAX assessment using clinical risk factors, and almost half (49%) had a sufficiently high probability of fracture to warrant further testing using a BMD-FRAX test. Yet, only 14% of the screened population had a resultant probability of fracture high enough to warrant intervention. This suggests that a method that improves screening efficiency and decreases the number of persons undergoing risk stratification, particularly BMD-FRAX assessments, would be a welcome addition to the screening strategy.

Skeletal measures that predict fracture risk are highly heritable (50%–85%) and include BMD and quantitative ultrasound speed of sound (SOS) measurements, which are highly correlated [10–13]. Recently, large cohort resources have enabled the genomic prediction of such heritable clinical risk factors from genotypes through polygenic risk scores [14–20], which capture information from many single nucleotide polymorphisms assayed from genome-wide genotyping. These assays assess common genetic variation at millions of single nucleotide polymorphisms and cost approximately $40 in a research context. However, the clinical utility of such polygenic risk scores is unclear, widespread replication of polygenic risk scores is currently lacking, and it is unknown whether they can aid in screening programs. Defining their clinical relevance may be particularly relevant in a British context, where the National Health Service aims to sequence 5 million individuals within 5 years [21].

Very large cohorts are required to train polygenic risk scores, and current cohorts lack sufficient sample size to generate useful BMD polygenic risk scores. However, since BMD is strongly correlated with SOS [22] and SOS has been measured in 341,449 individuals in UK Biobank, we developed a polygenic risk score for SOS termed "gSOS" (for "genetically predicted SOS") that could be used to identify individuals unlikely to have low enough BMD to warrant a clinical intervention. To improve screening efficiency, such individuals could be removed from an osteoporosis screening program prior to measurement of BMD. We then tested the generalizability and potential benefit of incorporating gSOS into the NOGG guidelines using 5 cohorts, comprising 10,522 eligible individuals. Last, we tested if gSOS could decrease the number of people requiring more detailed assessments, such as BMD measurement, while still identifying those who require interventions to decrease their risk of fracture.

## Methods

### Overall study design and cohorts

The purpose of this study was not to predict fractures. Rather, the purpose of this study was to understand if genetic prescreening could reduce the number of screening tests needed to identify individuals at risk of osteoporotic fractures. This study included 3 phases (Fig 1). The first 2 phases were conducted in 2 distinct subsets of the UK Biobank study cohort, and the final phase in a further subset of UK Biobank combined with 4 other cohorts. Characteristics of the cohorts are shown in Table 1, with the cohorts described in detail in Table A in S1 Tables.

The first phase used least absolute shrinkage and selection operator (LASSO) regression [23] to train a set of polygenic risk score models to predict SOS in the UK Biobank Training Set ($N = 341,449$). In phase 2, the polygenic risk score model explaining the most variance in measured SOS in the UK Biobank Model Selection Set ($N = 5,335$) was selected and named gSOS. The ability of gSOS to explain variance in measured SOS was then tested in the UK Biobank Test Set ($N = 84,768$). In phase 3, gSOS was tested for its performance in a screening strategy, based on NOGG guideline thresholds of fracture risk, applied to a population of 10,522 individuals derived from 5 separate cohorts. Inclusion in the screening program required these individuals to be $\geq$50 years with at least 1 risk factor and available measurement

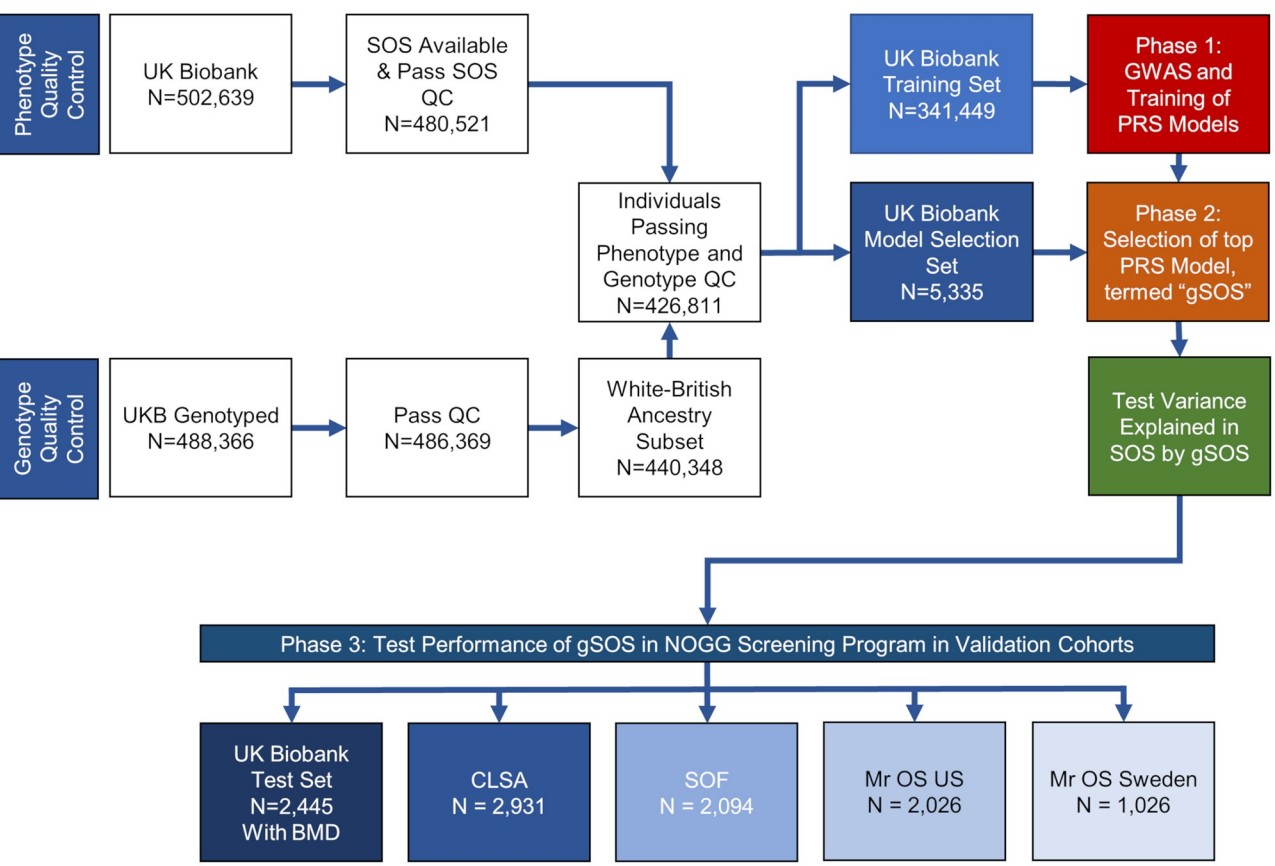

**Fig 1. Overall study design.** BMD, bone mineral density; CLSA, Canadian Longitudinal Study on Aging; GWAS, genome-wide association study; NOGG, National Osteoporosis Guideline Group; PRS, polygenic risk score; QC, quality control; SOF, Study of Osteoporotic Fractures; SOS, speed of sound; UKB, UK Biobank.

of femoral neck BMD. This population comprised a further distinct subset of the UK Biobank Test Set (*N* = 2,445), as well as individuals from the Canadian Longitudinal Study on Aging (CLSA) (*N* = 2,931), the Study of Osteoporotic Fractures (SOF) (*N* = 2,094), Mr OS US (2,026), and Mr OS Sweden (*N* = 1,026). Together these 5 cohorts in phase 3 are referred to as the validation cohorts. Next, to test the effect of gSOS on fracture screening by age, we stratified the CLSA cohort by age, dividing the population into 3 age groups: 50–59 years, 60–69 years, and ≥70 years. The CLSA cohort was chosen for this age-stratified analysis, because it was the largest validation cohort and had the widest age range. To assess the performance of gSOS in ancestries other than White British, we tested it in individuals in the UK Biobank Test Set who were eligible for screening and were of non–White British ancestry, as defined by genotypes (see S1 Text for further details of definition of ancestry; Table B in S1 Tables shows the demographic and risk factor characteristics of the sub-population).

This study adheres to the GRIPS statement (see S1 Checklist) and did not have a pre-specified analysis plan [24]. Specific ethics approval was not required for this study.

## SOS and BMD measurement

We decided to use polygenic risk scores to predict SOS, rather than BMD, because polygenic risk scores require a large number of individuals with proper phenotyping and genome-wide genotyping. The largest dataset for SOS is approximately 10-fold larger than that for BMD

**Table 1. Participant characteristics by dataset.**

| Participant characteristics | Model development cohorts | | gSOS-based screening test cohorts | | | | |
|---|---|---|---|---|---|---|---|
| | UK Biobank Training Set | UK Biobank Model Selection Set | UK Biobank Test Set | CLSA | SOF | Mr OS US | Mr OS Sweden |
| Sample size | 341,449 | 5,335 | 4,741 | 6,704 | 3,426 | 4,657 | 1,880 |
| Individuals eligible for screening, N (%) | — | — | 2,445 (51.6) | 2,931 (43.7) | 2,094 (61.1) | 2,026 (43.5) | 1,026 (54.6) |
| Age, mean (SD) | 56.8 (8.0) | 56.6 (8.1) | 55.8 (7.6) | 62.6 (9.9) | 71.5 (5.3) | 74.0 (6.0) | 75.4 (3.2) |
| Women, N (%) | 186,569 (55.6) | 2,863 (53.7) | 2,489 (52.5) | 3,396 (50.7) | 3,426 (100) | 0 (0) | 0 (0) |
| Smoker, N (%) | 27,181 (8.0) | 397 (7.4) | 966 (20.4) | 581 (8.7) | 270 (7.9) | 145 (3.1) | 178 (9.5) |
| Previous fracture, N (%) | 34,917 (10.2) | | 386 (8.1) | 1,032 (15.4) | 1,210 (35.3) | 1,084 (23.3) | 637 (33.9) |
| Use of glucocorticoids, N (%) | 3,330 (1.0) | 51 (0.8) | 79 (1.7) | 258 (3.9) | 363 (10.6) | 98 (2.1) | 34 (1.8) |
| Alcohol user, N (%) | — | — | — | 1,189 (17.7) | 98 (2.9) | 182 (3.9) | 52 (2.8) |
| Fall within last 12 months, N (%) | 69,057 (20.2) | 1,052 (20.0) | 1,500 (31.6) | 699 (10.4) | 1,021 (28.2) | 984 (21.1) | 298 (15.9) |
| Rheumatoid arthritis, N (%) | 3,312 (1.0) | 41 (0.8) | 28 (0.6) | 191 (2.9) | 252 (7.0) | 226 (4.9) | 27 (1.4) |
| Secondary osteoporosis, N (%) | 14,541 (4.3) | 215 (4.0) | 192 (4.1) | 313 (4.7) | — | — | — |
| Parental history of fracture, N (%) | — | — | — | 820 (12.2) | 404 (14.4) | 599 (16.8) | 164 (8.7) |
| Baseline CRF-FRAX score for MOF, mean (SD) | 5.1 (3.1) | 5.0 (3.1) | 4.8 (2.7) | 8.1 (6.8) | 18.7 (9.5) | 9.5 (4.7) | 11.1 (6.3) |
| Baseline BMD-FRAX score for MOF, mean (SD) | — | — | 4.9 (2.6) | 7.5 (5.8) | 17.1 (9.5) | 8.1 (4.4) | 13.1 (5.6) |
| gSOS, mean (SD) | — | −0.002 (1.00) | 0.043 (0.98) | −0.005 (1.00) | 0 (0.99) | −0.033 (0.98) | −0.708 (0.46) |

BMD-FRAX, bone-mineral-density-based Fracture Risk Assessment Tool; CLSA, Canadian Longitudinal Study on Aging; CRF-FRAX, clinical-risk-factor-based Fracture Risk Assessment Tool; MOF, major osteoporotic fracture; SOF, Study of Osteoporotic Fractures.

[10,25]. SOS also predicts fracture, with similar performance characteristics compared to BMD, and the 2 measures are correlated ($r = 0.4–0.6$) [22]. However, since femoral neck BMD is required for FRAX calculations used in screening programs [26], we required that all individuals in the phase 3 analysis have femoral neck BMD measure available. Details of SOS and BMD measurement are available in S1 Text. All analyses used SOS standardized to a mean of 0 and standard deviation of 1.

## Development of machine learning model to predict SOS

**Training, model selection, and test datasets.** To develop and test gSOS, we followed best practices in clinical prediction to ensure unbiased estimates of model performance by developing the models in datasets distinct from the datasets that were used to test model performance [27]. Participants in the UK Biobank with White British ancestry (see S1 Text), measured SOS, and genotyping information ($N = 426,811$) were randomly assigned to the UK Biobank Training Set (80% of participants), the UK Biobank Model Selection Set (1.25% of participants), or the UK Biobank Test Set (18.75% of participants) (Fig 1; Table 1). Since BMD was measured in only 4,741 individuals in all of UK Biobank [28], these individuals were assigned to the UK Biobank Test Set to enable them to be used in phase 3 of the study.

**Genome-wide association study (GWAS).** Using methods from our previous GWAS of estimated BMD in UK Biobank [25], but using a different sample size and SOS as the outcome, we undertook a GWAS for SOS in the UK Biobank Training Set ($N = 341,449$ individuals with

White British ancestry). We tested the additive allelic effects of each of the 13.9 million SNPs passing quality control, separately, on SOS using a linear mixed model to adjust for cryptic relatedness and population stratification [29], as well as adjusting for age, sex, assessment center, and genotyping array (S1 Text). Linkage-disequilibrium-independent associations where obtained using PLINK by clumping SNPs in linkage equilibrium at a $r^2 > 0.05$ and selecting a single most significant SNP from within each clumped set. To reduce potential bias due to population stratification, the UK Biobank Training, Model Selection, and Test Sets included only White British participants, while all other cohorts included only people of general European ancestry (as defined in S1 Text). Further, as stated above, the performance of gSOS-based screening was also tested in non–White British participants in UK Biobank.

**Polygenic risk scores using LASSO.** Using the UK Biobank Training Set, we fitted 6 LASSO models [23] to predict SOS using only SNPs with $p$-values smaller than a chosen set of thresholds (Table C in S1 Tables). The UK Biobank Model Selection Set was then used to identify the $p$-value threshold and regularization parameter ($\lambda$) that resulted in the lowest root mean square error for the prediction of SOS. This $p$-value threshold and regularization parameter were then taken forward for testing in the UK Biobank Test Set. Hence, we ensured that the performance of only 1 final polygenic risk score was evaluated in the UK Biobank Test Set. We refer to this final predictor as gSOS, in which SOS is predicted only from genotype.

**Traditional polygenic risk scores.** Traditional polygenic risk scores [15] were derived from the GWAS for SOS performed in the UK Biobank Training Set, without the use of LASSO, by including different sets of SNPs, selected by $p$-value threshold and linkage disequilibrium clumping as described in S1 Text (Table C in S1 Tables).

## Generation of FRAX scores

FRAX risk scores for major osteoporotic fracture (hip, clinical vertebra, proximal humerus, or wrist) can be generated with or without BMD, referred to in this paper as BMD-FRAX and CRF-FRAX, respectively [26]. Therefore CRF-FRAX and BMD-FRAX were calculated for all participants in each validation cohort [26]. FRAX clinical risk factors were assessed at the baseline visit for each cohort and included age, sex, body mass index (BMI), previous fracture, smoking, glucocorticoid use, rheumatoid arthritis, and secondary causes of osteoporosis. Measures of more than 2 daily units of alcohol and parental history of hip fracture were not available in UK Biobank and were set to "no" for this cohort, as suggested by FRAX guidelines. Not all secondary causes of osteoporosis were available for the SOF, Mr OS US, and Mr OS Sweden cohorts, and these variables were also set to "no" for these cohorts, as recommended by FRAX. Age was recorded at baseline visit. Sex was self-reported and verified by genotype. Individuals with discordant sex between self-report and genotype were excluded. CRF-FRAX and BMD-FRAX were calculated for all participants in each of the clinical cohorts, using country-specific FRAX models [26].

## Genomic screening in fracture risk screening

In the absence of an international consensus on fracture risk screening [2,4,5,30], we chose to use the assessment and management clinical algorithm developed by NOGG [3], since a screening program similar to the NOGG screening strategy is supported by randomized controlled trial evidence [9]. The NOGG screening strategy uses 10-year absolute probability of fracture as calculated by FRAX and suggests treatment or reassurance based on thresholds of risk, which are age dependent and consider competing risks. The NOGG guidelines (Fig 2) also aim to identify individuals at risk for fracture in a cost-efficient manner by reserving clinical visits and more costly BMD testing for those at intermediate risk, i.e., where the FRAX

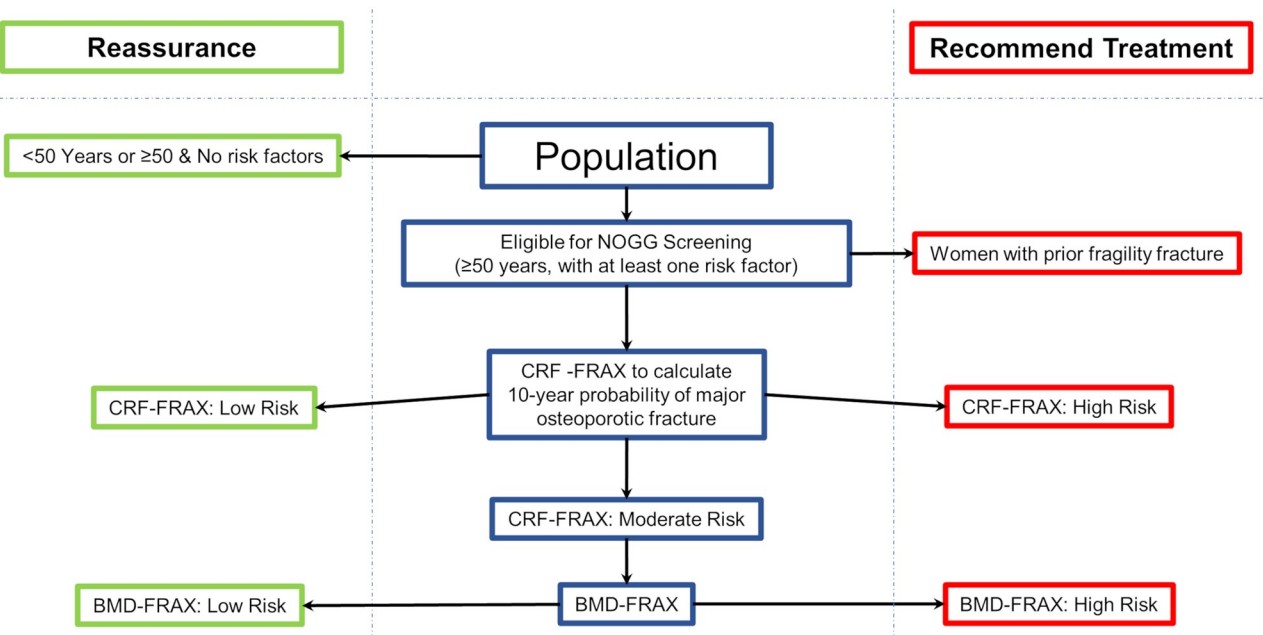

**Fig 2. NOGG screening strategy.** Both CRF-FRAX and BMD-FRAX generate a 10-year probability of major osteoporotic fracture, which is used to designate risk of fracture. BMD-FRAX, bone-mineral-density-based Fracture Risk Assessment Tool; CRF-FRAX, clinical-risk-factor-based Fracture Risk Assessment Tool; NOGG, National Osteoporosis Guideline Group.

score lies close to an intervention threshold. This intervention threshold is equivalent to the age-specific FRAX 10-year probability in women with a prior fragility fracture, since nearly all such women would be recommended an intervention [3]. Individuals without any risk factors are excluded from the CRF-FRAX assessment. By applying CRF-FRAX, individuals can be recommended for either an intervention (high risk), a BMD-FRAX assessment (intermediate risk), or reassurance and no further participation in the screening program (low risk). Those having a BMD-FRAX assessment can then be recommended an intervention if their resulting 10-year probability of major osteoporotic fracture exceeds the age-specific threshold, or they can be reassured (see Fig 2).

Despite the efficiencies gained by using this stepwise approach [31], false negatives can occur when interventions are not recommended to individuals who have a low CRF-FRAX-based probability and are discharged from subsequent screening, whereas if they had undergone BMD-FRAX, would have qualified for intervention. Likewise, false positives can arise when an individual is recommended for an intervention based on the CRF-FRAX score but would not have qualified for an intervention with BMD-FRAX.

To try to reduce the number of individuals undergoing testing, particularly more costly BMD testing, who would subsequently not require intervention, we introduced a gSOS-based screening step in the NOGG algorithm, where individuals were reassured if their gSOS was above a threshold (Fig 3). This is because individuals with a high SOS are likely to have a high BMD and are thus less likely to be recommended for an intervention. The trade-off of this strategy is that it could result in reassurance of individuals who, if their BMD was measured, would have been recommended an intervention. This would result in a decrease in sensitivity to identify individuals requiring an intervention. To calculate the sensitivity and specificity of the gSOS-modified NOGG algorithm, we used BMD-FRAX as a reference standard within the NOGG screening strategy (Fig 4). According to NOGG guidelines, women ≥50 years with a

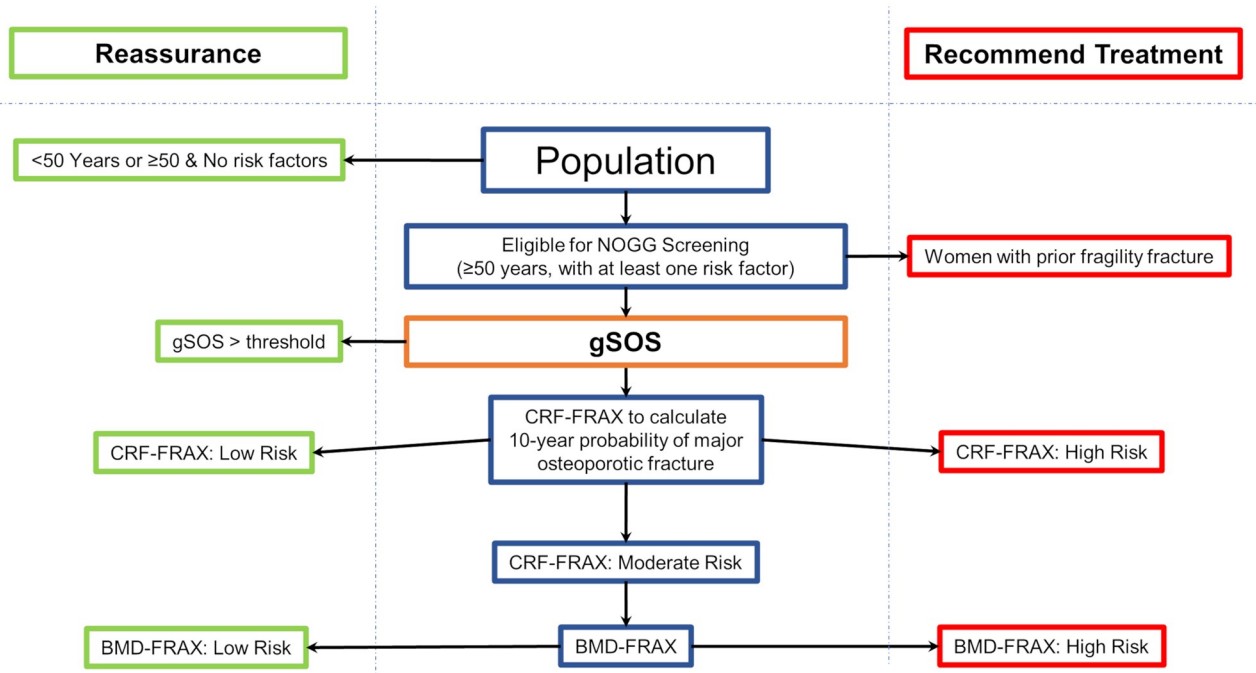

**Fig 3. NOGG screening strategy with a gSOS screening step.** Both CRF-based FRAX and BMD-based FRAX generate a 10-year probability of major osteoporotic fracture, which is used to designate risk of fracture. gSOS is standardized to have a mean of 0 and standard deviation of 1. BMD-FRAX, bone-mineral-density-based Fracture Risk Assessment Tool; CRF-FRAX, clinical-risk-factor-based Fracture Risk Assessment Tool; NOGG, National Osteoporosis Guideline Group.

prior fragility fracture are recommended treatment without further FRAX testing. As a result, these individuals were assigned an intervention recommendation when calculating the sensitivity and specificity of correct treatment assignment (Fig 4).

Since resources are often expended to measure BMD-FRAX in individuals whose final probability of fracture is too low to warrant intervention, we also estimated the number of

### BMD-FRAX based NOGG Guideline Result

| gSOS Modified NOGG Guideline Result | | Recommend Intervention | Reassurance |
|---|---|---|---|
| | Recommend Intervention | True Positives | False Positives |
| | Reassurance | False Negatives | True Negatives |

Sensitivity is derived from: True Positives / (True Positives + False Negatives)
Specificity is derived from: True Negatives / (True Negatives + False Positives)

**Fig 4. Calculation of sensitivity and specificity of correct treatment assignment.** BMD-FRAX, bone-mineral-density-based Fracture Risk Assessment Tool; NOGG, National Osteoporosis Guideline Group.

CRF-FRAX and BMD-FRAX tests that were performed but led to the individual being reassured without a recommended intervention.

We chose the sex-specific thresholds of gSOS that reduced CRF-FRAX and BMD-FRAX testing but minimized the loss of sensitivity to identify individuals who would be recommended for treatment. This threshold was chosen using data from the UK Biobank Test Set (S4 Fig). The generalizability of the selected gSOS threshold was then tested in the remaining 4 validation cohorts (CLSA, SOF, Mr OS US, and Mr OS Sweden). The number of CRF-FRAX and BMD-FRAX tests performed but not leading to an intervention were counted. These analyses were conducted in each validation cohort, men and women separately, and in all groups combined. We also tested individuals of non–White British ancestry in UK Biobank ($N = 350$), i.e., the individuals who remain subsequent to filtering out the White British subset and who have available measurements of femoral neck BMD. The characteristics are provided in Table B in S1 Tables.

## Results

### Cohort characteristics

Table 1 describes the FRAX risk factors for all of the cohorts. There were few clinically relevant differences in any of the osteoporosis-related risk factors in the UK Biobank Training, Model Selection, and Test Sets, as expected, since these sets were generated randomly. As planned, all individuals from UK Biobank with BMD measures were included in the UK Biobank Test Set, to ensure availability of BMD-FRAX scores as the reference standard. There were few differences in demographics or clinical risk factors between individuals with and without BMD measured. The validation cohorts (CLSA, SOF, Mr OS US, and Mr OS Sweden) provided a range of characteristics, allowing for a better assessment of the generalizability of results (Table 1).

### GWAS

After quality control (see S1 Text), 13,958,249 SNPs were included in the GWAS. The GWAS in the training set identified 1,404 independent ($r^2 \leq 0.05$) genome-wide significant loci at a p-value threshold of $<5 \times 10^{-8}$. S1 Fig shows the Manhattan and QQ plots for this GWAS.

### Variance explained in SOS in the UK Biobank Model Selection Set

The polygenic risk score models trained with LASSO explained at most 25.0% (95% CI 23.0%–27.0%) of the variance in SOS in the UK Biobank Model Selection Set (Table C in S1 Tables). S2 Fig provides detailed information on the optimal algorithm tuning parameters. None of the traditional polygenic risk scores performed better than the polygenic risk score derived from the LASSO regression. S3 Fig demonstrates that, as expected, the estimated effects of the activated SNPs from the LASSO algorithm were attenuated compared to the effects estimated from the GWAS.

### Variance explained in SOS in the UK Biobank Test Set

Age, sex, and BMI explained 4.0% (95% CI 3.7%–4.2%) of the variance in SOS. Adding all available FRAX clinical risk factors increased the variance explained to 5.3% (95% CI 5.0%–5.6%). The polygenic risk score from the UK Biobank Model Selection Set explaining the most variance in measured SOS was designated as "gSOS" and was then tested for its correlation with SOS in the UK Biobank Test Set. This model explained 23.2% (95% CI 22.7%–23.7%) of

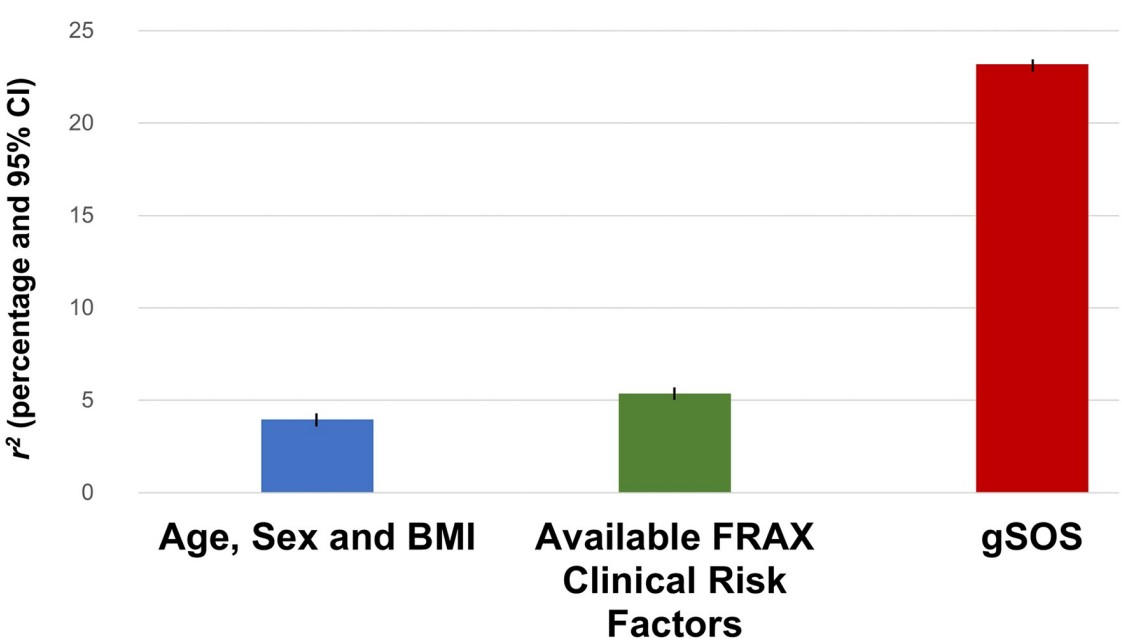

**Fig 5. Variance explained in SOS by clinical risk factors and gSOS in the UK Biobank Test Set.** Available FRAX clinical risk factors included age, sex, BMI, smoking, previous fracture, use of glucocorticoids, rheumatoid arthritis, and secondary osteoporosis. BMI, body mass index; FRAX, Fracture Risk Assessment Tool; SOS, speed of sound.

the variance in measured SOS and included 21,717 SNPs activated from a total of 345,111 SNPs that had $p$-values for association with SOS of $\leq 5 \times 10^{-4}$ (Table C in S1 Tables; Fig 5).

### Screening by NOGG guidelines in validation cohorts

The validation cohorts comprised 10,522 individuals eligible for fracture risk screening (Table 1). Both the sensitivity and specificity of the NOGG screening strategy to identify individuals at high enough risk to merit an intervention, compared to the reference standard, BMD-FRAX, were high (99.6% and 97.1%, respectively; Fig 6; Table D in S1 Tables). This high sensitivity and specificity required CRF-FRAX tests to be undertaken in 81% of the population eligible for screening, with BMD-FRAX tests subsequently recommended in 37% of the population. In total, 74% of those requiring CRF-FRAX tests were classified for reassurance, i.e., without a recommendation for an intervention. As well, just over one-third of all individuals who received a BMD-FRAX test were classified for reassurance without intervention (Fig 6; Table D in S1 Tables).

### Screening incorporating a gSOS-based screening step

Using the UK Biobank Test Set, we selected the threshold of gSOS that would minimize the number of BMD tests done in persons who would ultimately be reassured rather than receiving an intervention, but also would minimize the number of false negatives (S3 Fig). Applying this threshold separately in men and women, we found that a threshold of standardized gSOS set to 0.5 and 0 for men and women, respectively, balanced these goals in the UK Biobank Test Set, and subsequently individuals above these thresholds were excluded from further screening in the validation cohorts, prior to receiving a CRF-FRAX or BMD-FRAX test (Fig 3). The utility of this threshold was then tested in all validation cohorts.

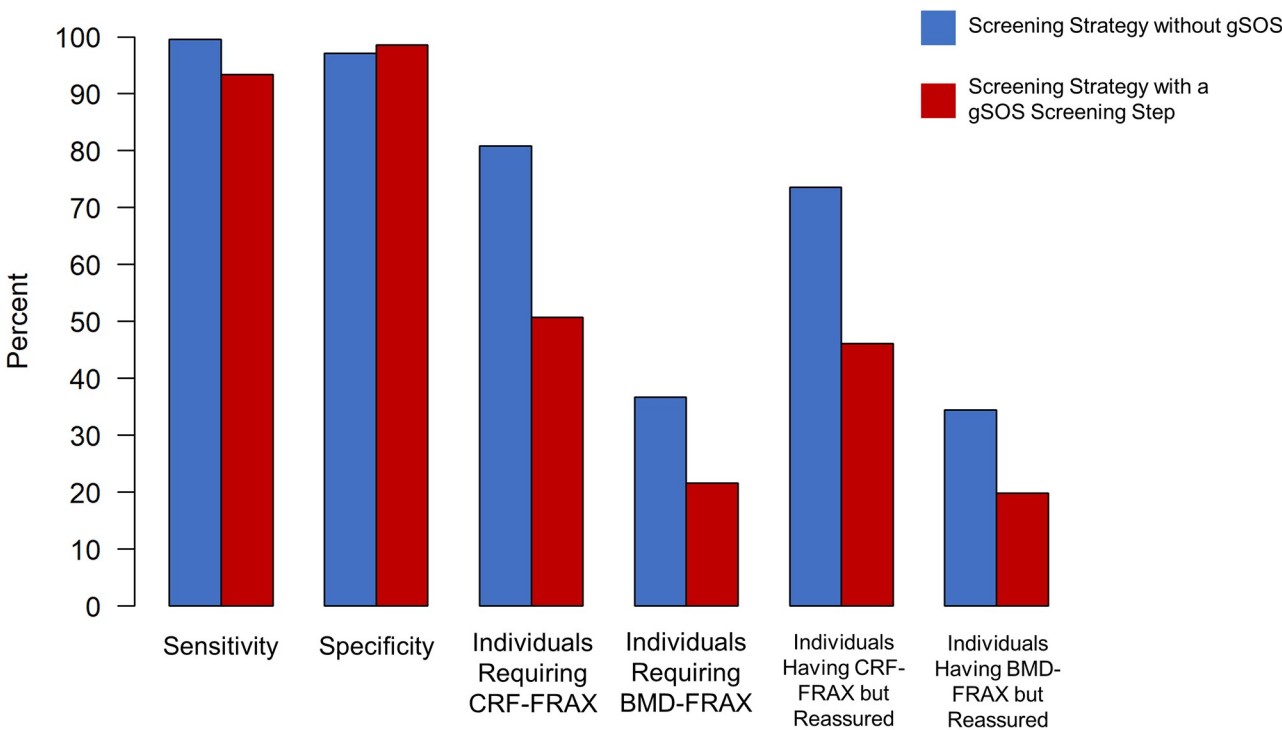

**Fig 6. Performance characteristics of screening with and without a gSOS screening step.** BMD-FRAX, bone-mineral-density-based Fracture Risk Assessment Tool; CRF-FRAX, clinical-risk-factor-based Fracture Risk Assessment Tool.

Fig 6 demonstrates that applying a gSOS screening step in the validation cohorts resulted in a small decrease in sensitivity to identify eligible participants for therapy, to 93.4%, but that the specificity increased slightly, to 98.5%. However, the proportion of screened individuals requiring CRF-FRAX testing decreased from 81% to 51% (representing a relative decrease of 37%) compared to NOGG-based screening without a gSOS screening step. Additionally, the proportion of screened individuals requiring BMD-FRAX testing decreased from 37% to 22% (representing a relative decrease of 41%) (Fig 6; Table D in S1 Tables).

The proportion of CRF-FRAX and BMD-FRAX tests that resulted in an individual being excluded from the screening program without a recommendation for an intervention also decreased from 74% to 46% and from 34% to 20%, respectively (Fig 6; Table D in S1 Tables). Cohort-specific results are shown in Tables E–I in S1 Tables.

The positive predictive value for correct treatment assignment in all validation cohorts was 91.8% without a gSOS screening step and increased to 95.4% with the gSOS screening step (Table D in S1 Tables; cohort-level results and subgroup results are available in Tables D–P in S1 Tables).

## Women and men separately

The SOF cohort was composed of only women, while Mr OS US and Mr OS Sweden were composed of only men, providing the opportunity to explore performance characteristics by sex. Further, we divided the UK Biobank Test Set and CLSA into sex-specific datasets (Tables J–M in S1 Tables). Amongst 4,859 women who were eligible for screening in the cohorts (SOF, UK Biobank Test Set, and CLSA), the sensitivity and specificity for correct treatment

assignment were high (99.9% and 95%, respectively). Nevertheless, 58% of the population required CRF-FRAX tests, and 43% required BMD-FRAX tests (Table N in S1 Tables).

When applying a gSOS screening step, the sensitivity decreased marginally, to 94.6%, and the specificity increased marginally, to 98.2%. The proportion of the population requiring a CRF-FRAX test reduced from 58% to 27% (representing a relative decrease of 55%), while the proportion requiring a BMD-FRAX test reduced from 43% to 20% (representing a relative decrease of 55%) (Table N in S1 Tables).

Amongst the 5,668 men eligible for screening, the sensitivity and specificity were 96.9% and 98.2%, respectively, using CRF-FRAX alone as the screening step. In order to achieve this performance, 100% of men had a CRF-FRAX test, and 31% required a BMD-FRAX test. The yield of high-risk individuals from these tests was low, such that 94% of men receiving a CRF-FRAX test were reassured, and 29% of those receiving a BMD-FRAX test were reassured (Table O in S1 Tables). Applying a gSOS screening step to these men reduced the sensitivity to 82% while maintaining a similar specificity at 99%. However, the proportion of men requiring a CRF-FRAX test reduced to 72% (representing a relative decrease of 28%), and the proportion undergoing BMD-FRAX reduced to 23% (representing a relative decrease of 25%).

## Stratification by age

We next tested the performance of gSOS in different age strata to understand if the screening efficiency improved more for one age group than another. Using the largest cohort, with the largest variation in age (CLSA, $N = 6,704$), we found that gSOS had the highest performance in individuals aged ≥70 years. Specifically, the sensitivity and specificity to identify individuals who require an intervention remained high, at 99.6% and 94.9%, respectively. The proportion of screened individuals requiring CRF-FRAX testing decreased from 73% to 37% (representing relative decrease of 49%) compared to the NOGG screening strategy without a gSOS screening step. Additionally, the proportion of screened individuals requiring BMD-FRAX testing decreased from 24% to 12% (representing a relative decrease of 50%) (Table F in S1 Tables). In contrast, in individuals aged 50–59 years, sensitivity reduced to 86%, but specificity was 99.6%. The percent of individuals requiring CRF-FRAX and BMD-FRAX testing reduced by 51% and 50%, respectively. This demonstrates that gSOS pre-screening improves the efficiency of screening, but that the sensitivity to correctly identify individuals requiring therapy is maximized in older age groups.

## Non–White British individuals

We then assessed the effect of a gSOS pre-screening in individuals from UK Biobank with dual-energy X-ray absorptiometry BMD measures who were of non–White British ancestry (Table B in S1 Tables). We found that the results were generally consistent with those in individuals of White British ancestry. Specifically, the proportion of screened individuals requiring CRF-FRAX testing decreased from 94% to 48% (representing a relative decrease of 49%) compared to NOGG-based screening without a gSOS screening step. Additionally, the proportion of screened individuals requiring BMD-FRAX testing decreased from 39% to 17% (representing a relative decrease of 57%) (Table P in S1 Tables).

The proportion of CRF-FRAX and BMD-FRAX tests that resulted in an individual being excluded from the screening program without a recommendation for an intervention also decreased from 92% to 47% and from 38% to 16%, respectively (Table P in S1 Tables).

## Discussion

By building a polygenic risk score using 341,449 individuals and validating its utility in fracture risk screening in 5 separate cohorts totaling 10,522 individuals, we determined that genomics-enabled fracture risk screening could reduce the proportion of people who require BMD-based testing by 41%, while maintaining a high overall sensitivity and specificity for correct treatment assignment. While these findings are not meant to be prescriptive, they indicate the possible utility of polygenic risk scores in screening programs that are dependent on heritable risk factors.

Fracture risk assessment is expensive, with estimates of approximately US$50,000 per quality-adjusted life year gained [32], but is less expensive, or even cost-saving, using NOGG assessment strategies [33,34], because NOGG decreases the number of individuals who require CRF-FRAX and BMD-FRAX testing. Current guidelines suggest testing a large proportion of the population [2,3,5], yet most screened individuals are not identified as having a clinically actionable level of fracture risk [9,35]. This circumstance provides an opportunity for genetically derived measures of risk to increase cost-efficiencies in healthcare systems where investments have been made in genome-wide genotyping. Already at least 7 large healthcare systems have invested in genome-wide genotyping of a large proportion of their population, among whom electronic health record data are available [36,37]. Since the costs associated with genome-wide genotyping have now dropped below those of several routine clinical tests, the use of polygenic risk scores could be particularly helpful in these environments since a one-time genotyping cost could be used to generate several polygenic risk scores. However, there is a clear need to study the translation of such polygenic risk scores to clinical applications [38]—and the work presented here provides one example of how such scores could be translated to the clinic.

Previous attempts to predict osteoporosis from genomic data did not substantially increase discrimination compared to standard clinical measures alone, likely because the GWAS that underpinned these attempts was derived from 32,961 individuals and explained only 5.8% of the variance in BMD [39,40]. The improvement in variance explained in this study was attributable to the increase in sample size afforded by UK Biobank and to the LASSO regression's ability learn SNP associations with SOS jointly, as opposed to summing over independently measured effects on BMD. Other attempts to predict BMD have been based on several dozen genome-wide significant SNPs [39], whereas our approach used machine learning to jointly consider the effects of 642,127 SNPs (Table C in S1 Tables). LASSO regression has recently been used to predict estimated BMD, but from a GWAS sample size that was one-third of that used here, explaining only 17.2% of the BMD variance, and it was not used in a screening program [14]. Our work has improved the genomic prediction of BMD and demonstrated its potential clinical relevance.

We observed similar predictive performance across all LASSO models in the model selection step (Table C in S1 Tables); therefore, it remains possible that a more parsimonious model containing fewer SNPs would perform as well. As a result, further exploration of these LASSO models is warranted in a future technical study. However, should a more complex model with more SNPs prove to be ideal, the hinderance to clinical translation should be minimal, as the computational burden is limited to the training of the models, and is not in the prediction of an individual's genetic risk.

The sensitivity and specificity to correctly assign intervention was maximized in individuals ≥70 years of age. This could be clinically relevant because this is the age range for which the SCOOP trial demonstrated that a community-based screening program could be effective in reducing hip fractures [9].

We acknowledge that for many practicing physicians, such as those in the UK, who have access to an automatically generated electronic-health-record-based CRF-FRAX test, the result of interest would be the reduction in BMD-FRAX tests. However, we observed no appreciable difference in the sensitivity and specificity to correctly identify individuals requiring therapy if the gSOS screening step was placed prior to the CRF-FRAX test or immediately after the CRF-FRAX test. Tables E–O in S1 Tables show the results for a reduction in BMD-FRAX tests by cohort and sex.

## Limitations

We have generated a polygenic risk score for SOS, rather than BMD, since there are insufficient data resources to generate such a score for BMD. Nevertheless, the correlation between SOS and BMD has enabled the identification of individuals unlikely to have a BMD low enough to warrant an intervention. Further refinement could improve the efficiencies presented here, including a polygenic risk score for BMD, when sample sizes are large enough to enable this. While nearly all FRAX risk factors were available for study, alcohol intake and parental history of fracture were not available from the UK Biobank cohorts. However, these were available in the other validation cohorts. Secondary causes of osteoporosis were not uniformly available in SOF, Mr OS US, and Mr OS Sweden. Nevertheless, CLSA provided similar results to other cohorts and had all required information. Like participants in most cohort studies, the participants used in these studies are, on average, healthier than the general population [41]. Thus, external validation in a truly population-based study may provide helpful estimates of the real-world performance of genomics-enabled fracture risk screening. While we have tested the utility of gSOS in individuals of non–White British ancestry, the sample size available for study was relatively small, and thus results should be replicated in additional cohorts of different ancestry, underlining the need for large-scale GWAS datasets in individuals of non-European ancestry [42]. We recognize that different approaches could be taken to incorporate polygenic risk scores into fracture risk screening, but here we offer a simple approach that could be readily implemented in a genotyped population with required FRAX risk factors using the NOGG strategy [9].

## Conclusions

In summary, we have developed and tested gSOS, a polygenic risk score for SOS, which when introduced into a fracture risk screening program decreased the number of people requiring CRF-FRAX and BMD-FRAX assessments, while still maintaining a high sensitivity and specificity to identify individuals in whom an intervention should be recommended. These findings highlight the role that genetic prediction could play in screening programs that rely upon heritable risk factors.

## Supporting information

**S1 Checklist. GRIPS Checklist.**
(DOCX)

**S1 Fig. GWAS of SOS.** (A) Manhattan plot from GWAS of SOS. (B) QQ plot from GWAS of SOS.
(TIFF)

**S2 Fig. Performance of each SNP set using LASSO regression in the model selection set.**
Each feature set consists of a set of SNPs associated with SOS at a specified *p*-value threshold (sub-panel titles). For each feature set, we fit a regularized model to the training set over a

range of regularization constants (λ) (top left), with each λ resulting in a variable subset of activated features (bottom left). The model with the minimal root mean square error in the model selection set (top right) was selected to compare the variance explained ($r^2$, bottom right among all feature sets.
(TIFF)

**S3 Fig. Correlation of effect estimates from the GWAS and the coefficients from the LASSO regression for activated SNPs.** Activated SNPs are those SNPs chosen by the machine learning algorithm to be in gSOS, the final selected model.
(TIFF)

**S4 Fig. Effects of gSOS threshold on treatment assignment.** Results stratified by women (top) and men (bottom).
(TIFF)

**S1 Tables. Supplemental tables.**
(XLSX)

**S1 Text. Supplemental text.**
(DOCX)

## Acknowledgments

This research has been conducted using the UK Biobank Resource under project number 24268. We appreciate the generosity of UK Biobank and validation cohort volunteers. We appreciate advice on the manuscript provided by Dr. Suzanne Morin.

## Author Contributions

**Conceptualization:** Vincenzo Forgetta, Julyan Keller-Baruch, Marie Forest, Audrey Durand, Sahir Bhatnagar, John A. Morris, Eugene V. McCloskey, Claes Ohlsson, Joelle Pineau, William D. Leslie, Celia M. T. Greenwood, J. Brent Richards.

**Data curation:** Vincenzo Forgetta, Julyan Keller-Baruch, Marie Forest.

**Formal analysis:** Vincenzo Forgetta, Julyan Keller-Baruch, Marie Forest, Audrey Durand, Sahir Bhatnagar, John P. Kemp, Maria Nethander, J. Brent Richards.

**Funding acquisition:** J. Brent Richards.

**Investigation:** Vincenzo Forgetta, Julyan Keller-Baruch, J. Brent Richards.

**Methodology:** Vincenzo Forgetta, Julyan Keller-Baruch, Marie Forest, Audrey Durand, Sahir Bhatnagar, Joelle Pineau, Celia M. T. Greenwood.

**Resources:** John P. Kemp, John A. Morris, David M. Evans.

**Software:** Vincenzo Forgetta, Julyan Keller-Baruch, Marie Forest, Audrey Durand, Sahir Bhatnagar.

**Supervision:** J. Brent Richards.

**Validation:** Maria Nethander, Daniel Evans, Douglas P. Kiel, Fernando Rivadeneira, Helena Johansson, Nicholas C. Harvey, Dan Mellström, Magnus Karlsson, Cyrus Cooper, David M. Evans, Robert Clarke, John A. Kanis, Eric Orwoll, Claes Ohlsson.

**Visualization:** Vincenzo Forgetta, Julyan Keller-Baruch.

**Writing – original draft:** Dan Mellström, J. Brent Richards.

**Writing – review & editing:** Vincenzo Forgetta, Julyan Keller-Baruch, Marie Forest, Audrey Durand, Sahir Bhatnagar, John P. Kemp, Maria Nethander, Daniel Evans, John A. Morris, Douglas P. Kiel, Fernando Rivadeneira, Helena Johansson, Nicholas C. Harvey, Magnus Karlsson, Cyrus Cooper, David M. Evans, Robert Clarke, John A. Kanis, Eric Orwoll, Eugene V. McCloskey, Claes Ohlsson, Joelle Pineau, William D. Leslie, Celia M. T. Greenwood, J. Brent Richards.

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
