## [Decision Letter · Decision Letter 0]

4 Feb 2020

Dear Dr. Richards,

Thank you very much for submitting your manuscript "A Polygenic Risk Score to Improve Screening for Fracture Risk" (PMEDICINE-D-19-04448) for consideration at PLOS Medicine. 

Your paper was discussed among the editorial team, evaluated by an academic editor with relevant expertise, and sent to independent reviewers, including a statistical reviewer. The reviews are appended at the bottom of this email and any accompanying reviewer attachments can be seen via the link below:

[LINK]

In light of these reviews, we will not be able to accept the manuscript for publication in the journal in its current form, but we would like to invite you to submit a revised version that fully addresses the reviewers' and editors' comments. You will appreciate that we cannot make a decision about publication until we have seen the revised manuscript and your response, and we expect to seek re-review by one or more of the reviewers. 

We hope to receive your revised manuscript by Feb 25 2020 11:59PM. Please email us (plosmedicine@plos.org) if you have any questions or concerns.

Please let me know if you have any questions. Otherwise, we look forward to receiving your revised manuscript. 

Sincerely,

Richard Turner PhD, for Louise Gaynor-Brook, MBBS PhD

Associate Editor, PLOS Medicine

rturner@plos.org

Throughout your text, please adapt the wording to reflect the experimental design rather than anticipations of possible use dependent on further clinical studies. For example, "The use of a polygenic risk score ... can reduce ..." seems too stridently worded. 

Please add a study descriptor to the title. We suggest: "Development of a polygenic risk score to improve screening for fracture risk: a cross-sectional study". 

Please convert the abstract to PLOS Medicine style, i.e., with three subsections. The final sentence of the "methods and findings" subsection should summarize the study's main limitations. 

Please begin the "Conclusions" subsection of your abstract with "In this study, we found that ..." or similar. 

Please quote summary demographic and clinical details for study participants in your abstract. 

In your abstract and elsewhere in the paper, please add p values alongside 95% CI where available. 

After your abstract, we will need to ask you to add a new and accessible "author summary" section in non-identical prose. You may find it helpful to consult one or two recently published research papers in PLOS Medicine to get a sense of the preferred style. 

Early in your methods section, please state whether the study had a protocol or prespecified analysis plan and if so attach the relevant document(s) as a supplementary file (referred to in the text). Please highlight analyses that were not prespecified. 

Please remove trademarks from the manuscript.

Throughout the text, please format reference call-outs as follows: "... that for BMD [10,24]."

Please ensure that citations in your reference list meet journal style. All boldface and italic elements should be converted into plain text. Where appropriate, 6 author names should be listed, followed by "et al.".

Please add a journal name to reference 37 as appropriate. 

If references 41 and 43 are preprints, please indicate this. 

Please add a completed checklist for the most appropriate reporting guideline, which may be STROBE, referred to in your methods section. In the checklist, individual items should be referred to by section (e.g., "Methods") and paragraph number rather than by page or line numbers, as the latter generally change in the event of publication. 

Comments from the reviewers:

*** Reviewer #1: 

The authors present a well-written and interesting study on developing a polygenic risk score for heal quantitative ultrasound speed of sound (SOS) for identify low risk individuals who can excluded from fracture risk screening. The strengths of the study include, developing and testing the genetic risk score (GRS) in a large population cohort (UK Biobank), interestingly using both a LASSO-based data-driven approach, and then validating the performance of the GRS in 5 other cohorts with FRAX-score risk factors available , including Bone Mineral Density. The key finding being that there were substantial reductions in proportion of individuals requiring FRAX-based tests (with or without BMD).

There are quite a few analyses in this and some of the results are difficult to follow as they are given in excel workbooks instead of word format. However, the statistical methods appear sound - with appropriate QC of genotyping data and SNP selection. The introduction of the LASSO model compared to simple polygenic risk score is also a nice addition and showed that using a data-driven approach can improve the variance in the model (up to 25%). 

Much of the literature on polygenic risk scores has been done in the context of "how much added values does GRS have above and beyond a clinical prediction model" with a focus on the overall change in AUC. The authors have framed their research question differently, in particular about how a GRS could preclude people from undertaking FRAX risk assessment (which have some level of costs). Whilst the modelling is convincing comparing these strategies - I am still left wondering about the clinical utility and feasibility of such an approach given that genotyping will need to be done on ALL individuals. Albeit the authors mentioned genotyping is more available now but that is really limited still to research based studies - it not a strategy which is indeed offered as routine clinical care. 

Specific comments:

(1) The project clearly shows the potential of using polygenic risk score but it would be useful if the authors could comment about the clinical utility of their approach. There is a growing drive for genomic testing in routine practice but given that their LASSO model included 21,717 activated SNPs mean that's genotyping arrays of this magnitude will need to be utilised - which will indeed a barrier for implementation. The authors themselves had to use Amazon cloud computing to conduct the analyses. I'm wondering if a reduced set of SNPs would confer similar set of results. In Table S2, the LASSO approach with 6,823 SNPS explains about 23.4% which is not far off the 24.99% achieved by having 21,717 SNPs. There is in fact an argument for using more simplistic GRS scores with fewer SNPS. 

(2) Please check text results section (Lines 344) as it says the SNP array with 21,717 activated SNPS has a r^2 of 23.2% but in Table S2 the r^2 for the 21,717 SNPS

(3) CRF-FRAX is significantly cheaper to apply without BMD - in fact many EHRs have this embedded within medical record. In the UK for instance, TPP offers CRF-FRAX and EMIS offers QFracture using data which already collected in patient's medical records. There is logical mismatch in argument made by the authors in the discussion on "requiring all individuals to undergo a CRF-FRAX test" as the scores are CRF-FRAX or QFractures scores are estimated instantaneously from the medical record. The reductions in individuals requiring CRF-FRAX is actually not a huge burden with algorithms being integrated into EHRs. The more compelling argument here is the reduction of BMD-based FRAX tests. 

(4) I found it interesting the authors looked at placing the genomic screening step after the CRF-FRAX and similar to other studies which have looked at the added value of the GRS on top of the clinical prediction model, they found no further improvement in sensitivity and specificity. As a population screening model the key comparison should be the CRF-FRAX model as this has been embedded in primary care EHRs at very minimal costs. The current comparison is BMD-FRAX but the more commonly and cheaper alternate which is adopted in primary care is CRF-FRAX due to BMD not being routinely available. 

*** Reviewer #2: 

Forgetta et al., 2019

Forgetta et al. analyze data from UK Biobank to derive a polygenic risk score (gSOS) for heel ultrasound speed of sound. They show that the polygenic risk score can be used to reduce the number of patients that require DXA scans for bone mineral density in the assessment of fracture risk. 

This paper tests gSOS in clinical cohorts. The authors show that gSOS can be used to eliminate the need for DXA scans for some patients, thereby decreasing the cost of screening a general population. The authors set a threshold for gSOS such that patients above the threshold are "reassured" and do not require further screening. The key result can be seen in Fig. 6, which shows that incorporating gSOS retains sensitivity and specificity but decreases the number of patients that require CRF-FRAX and BMD-FRX. The figure clearly shows the cost savings of genetic testing in diagnosing osteoporosis. 

I reviewed a previous version of this paper, and the current submission has already responded to many of my comments. 

Minor comments

Line 441 "Recently, LASSO regression was used to predict estimated BMD, but from a GWAS sample size that was one third of that used here, captured only 20% of the its variance and was not used in a screening program.41" This statement should refer to reference 14, which developed gSOS previously based on the identical UK Biobank cohort and with nearly identical performance (i.e. correlation to SOS) as in this submitted work. 

In Table 1, some of the demographics differ between groups and should have comments:

1. The percent smoker in the Biobank test set is much different than the training or selection set. Smoker is a key factor for FRAX, and this difference could impact performance of gSOS. 

2. Age in the four clinical cohorts is much higher than the Biobank study.

3. The percent of previous fracture in the four clinical cohorts is higher than in the UK Biobank. Previous fracture is a very strong risk factor for osteoporotic fractures. This difference likely impacts the results. 

4. The percent of falls in the Biobank test set was higher than in the training or marker selection set. This is very worrisome as it suggests that the test set was at higher risk for fragility fracture than the training set.

Table 1 should list 95% CI for each of the percentages to show which of the difference are statistically significant.

*** Reviewer #3: 

Forgetta et al. have developed and tested a polygenic prediction tool for bone fracture, based on the large UK Biobank sample (341,449 individuals) and a polygenic risk score for heel quantitative ultrasound speed of sound (SOS). The paper is timely and interesting, and shows that genetic testing may reduce the need for additional screening tests. Some issues should be addressed: 

1) The use case can be improved. The risk for an event is less clinically relevant than prediction the time of the event. It would be of interest to see the results with for example the polygenic hazard score in this context (published in PLOS Medicine in 2017). 

2) Further, in what age groups are the gSOS most predictive? This should be added to the model. 

3) What is the positive predictive value of gSOS? These result should be presented. 

4) Outcome: the sensitivity and specificity did not improve, while the proportion of individuals requiring CRF-FRAX testing decreased from 81% to 51% and the proportion of screened individuals requiring BMD-FRAX testing decreased from 37% to 22%. However, this seems to be age dependent, and for which age group is this relevant? 

5) The sensitivity and specificity of gSOS is not better than the for the other measures (a small decrease in sensitivity to 93.4%, while the specificity increased slightly to 98.5%). This result should be discussed, and the reason for lack of improvement should be presented. 

6) Earlier intervention is often better for prevention. Does the gSOS improve the predictive value of BMD-FRAX or CRF-FRAX if applied to younger age groups? This is a highly relevant analysis. 

7) UK Biobank - the GWAS did not include age as variable? Is it possible to provide result of age dependent GWAS, at least for different age groups? 

8) There should be non-European samples available that the authors can use for testing the performance in other ancestries.

***

[LINK]

---

## [Decision Letter · Decision Letter 1]

19 Apr 2020

Dear Dr. Richards,

Thank you very much for submitting your revised manuscript 'Development of a polygenic risk score to improve screening for fracture risk' (PMEDICINE-D-19-04448R1) for consideration at PLOS Medicine. 

Your paper was discussed with our academic editor and was also seen by one peer reviewer, whose comments you can read at the bottom of this email. Any accompanying reviewer attachments can be seen via the link below:

[LINK]

We would now like to invite you to submit a further revised version that fully addresses the reviewer's and editors' comments. Please note that we cannot make a decision about publication until we have seen the revised manuscript and your response, and we may seek re-review by one or more reviewers. 

In revising the manuscript for further consideration here, your revisions should address the specific points made by each reviewer and the editors. Please also check the guidelines for revised papers at http://journals.plos.org/plosmedicine/s/revising-your-manuscript for any that apply to your paper. In your rebuttal letter you should indicate your response to any comments from reviewers or editors and the changes you have made in the manuscript. Please submit a clean version of the paper as the main article file; a version with changes marked should be uploaded as a marked up manuscript. 

In addition, we request that you upload any figures associated with your paper as individual TIF or EPS files with 300dpi resolution at resubmission; please read our figure guidelines for more information on our requirements: http://journals.plos.org/plosmedicine/s/figures.

While revising your submission, please upload your figure files to the PACE digital diagnostic tool, https://pace.apexcovantage.com/ PACE helps ensure that figures meet PLOS requirements. To use PACE, you must first register as a user. Then, login and navigate to the UPLOAD tab, where you will find detailed instructions on how to use the tool. If you encounter any issues or have any questions when using PACE, please email us at PLOSMedicine@plos.org.

We hope to receive your revised manuscript by May 08 2020 11:59PM. Please email us (plosmedicine@plos.org) if you have any questions or concerns.

Your article can be found in the 'Submissions Needing Revision' folder. 

Please let me know if you have any questions. Otherwise, we look forward to receiving your revised manuscript shortly. 

Sincerely,

Richard Turner PhD

Senior editor, PLOS Medicine

rturner@plos.org

To your data statement, please add the UK Biobank contact details for those wishing to inquire about access to data.

Please add a study descriptor to your title, following a colon, e.g. "...: a genetic risk prediction study". 

Please add a new concluding sentence to the "methods and findings" subsection of your abstract, quoting 2-3 of the study's main limitations. 

If not done, please add a sentence to the methods section to note that the study did not have a pre-specified analysis plan. 

Throughout the paper, please format reference call-outs as follows, preceding punctuation: "... [1].".

Please add p values alongside 95% CI, where available. 

Please remove all trade marks, e.g., at line 67.

Please present the reporting checklist in a separate supplementary document, referred to in your methods section. 

Comments from the reviewers:

*** Reviewer #3: 

The authors have addressed my comments adequately, except for the points about prediction which seems to be based on a misunderstanding. 

I understand they are not predicting fracture. However, they are predicting an event: who requires an intervention ("be eligible for an intervention", "identify individuals requiring treatment".)

They are estimating the sensitivity and specificity for this event. 

Thus, they should be able to provide

1) Estimate the time to the event (requiring treatment), or this is a limitation

2) The age group where the sensitivity and specificity is highest for the event (to identify individuals requiring treatment), 

3) The positive predictive value for the event (requiring treatment)

They should provide standard clinical measures for prediction performance to identify individuals who require an intervention based on their quantitative measure. Otherwise, the approach has no clinical interest

***

[LINK]

---

## [Decision Letter · Decision Letter 2]

8 May 2020

Dear Dr. Richards,

Thank you very much for re-submitting your manuscript "Development of a polygenic risk score to improve screening for fracture risk: a genetic risk prediction study" (PMEDICINE-D-19-04448R2) for consideration at PLOS Medicine.

We have discussed the paper with our academic editor and it was also seen again by one reviewer. I am pleased to tell you that, provided the remaining editorial and production issues are fully dealt with, we expect to be able to accept the paper for publication in the journal.

[LINK]

Please let me know if you have any questions. Otherwise we look forward to receiving the revised manuscript shortly. 

Sincerely,

Richard Turner PhD

Senior editor, PLOS Medicine

rturner@plos.org

Requests from Editors:

Please truncate the short title. 

Thank you for adding an additional sentence at the end of the "methods and findings" subsection of your abstract. However, journal style calls for a sentence that summarizes limitations of the present study rather than detailing future research plans. Please substitute a sentence such as "Study limitations include a reliance on cohorts of predominantly European ethnicity, and use of a proxy of fracture risk.".

Our academic editor commented that, in view of the in silico research design, the language used regarding implied clinical utility would need to be amended prior to publication. For example, at line 34 rather than "... we found that the use of ... decreased ...", more apt wording would be "... our results suggest that the use of ... could decrease the number of individuals ...". Similarly, at line 50, the wording should be changed to "... we estimate that reassuring individuals ... could reduce ...". 

Please review the entire manuscript and make similar changes (e.g., at lines 386 the text could be amended to "... screening could reduce the proportion ..."; and at line 388 "... they indicate the possible utility of polygenic risk scores...").

Around line 145, please add "Specific ethics approval was not required for this study" or similar. 

Please make that "characteristics are" at line 264. 

Please adapt the formatting of the reference call-outs throughout the text ("... are available [37,38]. Since the ...").

Please add full author and access details to reference 10. 

Please break the completed GRIPS checklist out into a separate supplementary file, referred to in your methods section as "See S1_GRIPS_Checklist" or similar. 

Comments from Reviewers:

*** Reviewer #3: 

The remaining issues have now been addressed

***

[LINK]

---

## [Editor Report · Decision Letter 3]

3 Jun 2020

Dear Dr. Richards, 

On behalf of my colleagues and the academic editor, Dr. Christelle Nguyen, I am delighted to inform you that your manuscript entitled "Development of a polygenic risk score to improve screening for fracture risk: a genetic risk prediction study" (PMEDICINE-D-19-04448R3) has been accepted for publication in PLOS Medicine. 

PRODUCTION PROCESS

PRESS

PROFILE INFORMATION

Thank you again for submitting the manuscript to PLOS Medicine. We look forward to publishing it. 

Best wishes, 

Richard Turner, PhD

Senior Editor 

PLOS Medicine

plosmedicine.org